# Comparison of Pericarp Functional Traits in *Capparis spinosa* from Coastal and Inland Mediterranean Habitats

**DOI:** 10.3390/plants11223085

**Published:** 2022-11-14

**Authors:** Savvas Christodoulou, Chrysanthi Chimona, Sophia Rhizopoulou

**Affiliations:** Section of Botany, Department of Biology, National and Kapodistrian University of Athens, 15784 Athens, Greece

**Keywords:** caper, carbohydrates, drought, latitude, nitrogen, pericarp, proline, summer

## Abstract

The caper (*Capparis spinosa* L.) is a winter deciduous, perennial plant that grows and completes its life cycle entirely during the dry season in the Mediterranean region. Mature caper fruits and their pericarp, collected from the wild shrubs of the *Capparis spinosa* grown in the inland and coastal sites of Greece during summer, have been studied in order to improve and complete our knowledge of the successful establishment of the *C. spinosa* in Mediterranean ecosystems. Caper fruits possess substantial nutritional, medicinal and ecological properties that vary according to the developmental stage, agroclimatic and geographical parameters; however, the fruit pericarp and pedicel, unlike the other aboveground plant parts of the caper, have not hitherto been studied. The higher sugar and starch content in the pericarps and fruit pedicels harvested from wild caper plants grown in coastal habitats was investigated in comparison with those from inland habitats, while the higher proline and nitrogen content in pericarps and fruit pedicels harvested from wild caper plants grown in inland habitats was investigated in comparison with those from coastal habitats. The PCA, based on the considered functional traits underlying the constitutional aspects, reveals groupings of fruit pericarp specimens of the *C. spinosa* collected from coastal and inland habitats that are grounds for adaptive variation.

## 1. Introduction

The perennial, branched bush *Capparis* L. (*Species Plantarum* 1:503, 1753) is widely distributed in arid and semi-arid landscapes [1,2,3]; in Dioscoridis’ *De Materia Medica* (c. 64 AD) it is quoted as a plant distinct enough not to be confused with anything else, being referred to as “κάππαρις” (kapparis) [4,5,6]. The life cycle of the winter-deciduous *Capparis spinosa* L. (caper) lasts for six months, i.e., from May to October in the eastern Mediterranean [7,8,9]. *C. spinosa* begins to grow in May by forming new green stems that grow and branch close to the soil surface, as well as in crevices creeping along steep rocky cliffs, stony slopes, ruins and archaeological sites that act as conservative habitats [6,10,11,12,13,14,15]. The rapidly expanding leaves and numerous flower buds (Figure 1A) are developed on new elongating stems. *C. spinosa* blossoms entirely during the summer dry period, which is not favorable for flowering in the eastern Mediterranean [16,17]; its flowers (Figure 1B) expand at dusk and lose their turgor soon after the oncoming sunrise [17,18]. Such a rapid growth requires a substantial supply of water via a deep, extensive and conductive root system, which is another striking feature of this species [13,19,20,21].

Numerous studies in recent years have focused on the function, structure, chemical composition and medicinal uses of the flowers, buds, thorny stems, leaves and fruits of the *C. spinosa*, as well as this species’ evolution and the dispersal of its numerous seeds from dehiscent mature fruits [18,22,23,24,25,26,27,28,29,30,31,32,33,34,35,36]. It has been reported that wasps, which are active concomitantly with the ripening of caper fruits, act as “vectors” of its seeds, enhancing their dispersal [37]. Also, the odor [38,39,40] and other substances of mature caper fruits attract ants and lizards that serve as vehicles, influencing the long-distance dispersal of its small seeds that are adhered to their bodies and transferred to their nests [41,42].

The aboveground plant parts of the *C. spinosa* (Figure 1) are diurnally exposed to elevated air temperature (even above 30 °C) and negligible precipitation during the dry summer period; however, *C. spinosa* is not subjected to any water shortage, because its extensive and elongated root system detects water in deep soil layers, thus maintaining increased photosynthetic rate and stomatal conductance during the drought season [13,20,22,23]. The salient features of the growth of *C. spinosa* demonstrate a sophisticated plant response to drought, involving osmotic adjustment, cell wall properties and increased root density [20]. In addition, the regulated stomatal opening (throughout the photoperiod) sustains the elevated transpiration rates that induce the cooling of the exposed leaf surfaces of *C. spinosa* [8,21,22]. The impressive flowers of the caper open at dusk via a rapid petal extension, which involves the presence of expansins and the maintenance of elevated turgor in petals, which is also modified by the contribution of soluble sugars and proline [18,25,43,44]. Furthermore, it has been argued that insects’ visitation is synchronized to soluble sugar accumulation in the exudated nectar by floral tissues [45,46].

The pericarp—etymologically derived from two Greek words, i.e., peri (περί): around and carpos (καρπός): fruit—is developed from the ovary wall and is a portion of the fruit that may consist of three layers [47], more or less distinct morphologically, i.e., the epicarp or exocarp that is the outermost layer, the mesocarp that is the median layer and the inmost pulpy endocarp with the numerous, adhered seeds of *C. spinosa*; this is probably linked to the homogalacturonan found in the fruits of *C. spinosa* [48,49]. However, in the thin pericarp of the fleshy fruits the three layers merge and it is very difficult to distinguish them from one another. The accumulated substances in the pericarp provide resistance to disease and pests, and contribute to the fruit’s flavor and color [50]. The caper fruits, surrounded by the pericarp, possess substantial nutritional, medicinal and ecological properties that vary according to their developmental stage, climatic conditions and geographical locations [30,51,52,53].

The objective of this work was to study and compare the soluble sugar, starch, nitrogen and proline content of the pericarp and pedicel of wild *C. spinosa* grown under ambient conditions in coastal and inland habitats of Greece and exposed to the severity of the dry season. Actually, our research interest was triggered by the rapid fruit maturation and functionality of *C. spinosa*, which depend on photosynthate resources and water availability, during the summer drought period. In addition, the study of the considered functional traits of the pericarp and pedicel was linked with our previous knowledge of tissues from plants of *C. spinosa* growing in the wild [7,8,13,17,18,44]. According to the best of our knowledge the pericarp, unlike the other aboveground plant parts of the caper that have attracted considerable research, has not hitherto been studied.

## 2. Results

### 2.1. Phenological Stages

Τhe phenological pattern of the perennial, winter deciduous *C. spinosa* set within the context of the seasons includes new stem and leaf emergence in late spring and their senescence in autumn, flowering and fruiting entirely during the dry summer in the eastern Mediterranean (Figure 2). Therefore, *C. spinosa* has a five-month carbon-gaining period, a two-month period of flowering and one-month of fruiting, which indicates the potential of this species for resource acquisition.

### 2.2. Soluble Sugars

The soluble sugar content of the pericarp of the caper fluctuated among the considered sites; a two-fold to four-fold higher soluble sugar content was estimated in the samples from the coastal sites, in comparison with that detected in the samples from the inland sites (Figure 3). The highest sugar content (658 mg g^−1^) was detected in the coastal site V (23.7736° E, 37.7983° N), which was not significantly higher than that in the other coastal cases, while the lowest sugar content (85 mg g^−1^) in the inland site EK (23.7870° E, 37.9664° N) was comparable with those from the other inland cases. Along an increased latitude gradient, the pericarp soluble sugar content increased in coastal specimens (with the exception of those from E site: 23.4432° E, 38.8841° N), while a slight decline was detected in specimens from the EK inland site.

### 2.3. Starch

The starch content of the caper pericarp from samples of the inland and coastal sites is illustrated in Figure 4. The highest values of starch content of the caper pericarp were obtained in samples from the S (24.6607° E, 37.0612° N) and E (23.4432° E, 38.8841° N) coastal sites, while the lowest value (72 mg g^−1^) was detected in samples from the Ky (21.6779° E, 37.2488° N) inland site (Figure 4).

### 2.4. Nitrogen

Similar values of nitrogen content (Figure 5) were detected in the pericarp of the caper fruits collected from the inland sites despite the increased latitude. The nitrogen content of samples from the coastal sites declined with increased latitude (Figure 5), with the exception of the Cy site (22.9904° E, 36.2633° N) where the nitrogen content was significantly higher than that of the other coastal sites (*p* < 0.05).

### 2.5. Free Proline Accumulation

The high and low values of free proline accumulation in the pericarp of the caper fruits from the inland and coastal sites are presented in Figure 6; the proline content of the caper pericarp from the samples of the coastal sites, with the exception of the Cy coastal site (Figure 6), was lower (*p* < 0.05) than that of the inland sites.

### 2.6. Principal Component Analysis

The Principal Component Analysis (PCA) (Figure 7) shows that PC1 and PC2 reveal significant (99.9%) differentiation between coastal and inland caper pericarps. The considered ecophysiological traits of pericarps indicate a substantial grouping of the fruit specimens collected from the wild caper plants from coastal versus inland habitats.

### 2.7. Pedicels

In Table 1, the results of the soluble sugar, starch, nitrogen and proline content of the pedicels are presented. A significant positive correlation was detected between pericarps and pedicels in the cases of soluble sugars (Spearman coefficient: r_s_ = 0.84) and proline (r_s_ = 0.82). A non-significant positive correlation was detected between pericarps and pedicels in the case of starch (r_s_ = 0.31), while there was a negative correlation in the case of nitrogen (r_s_ = −0.41). Also, a negative relationship (*p* < 0.05) was detected between the soluble sugars of pericarps and the starch of pedicels (r_s_ = −0.63).

## 3. Discussion

The fruit of the *C. spinosa* grows and completes its development within a month during the summer drought period in the considered research sites. The fruit (caperberry) is ellipsoid with a thin pericarp and a large quantity of seeds [54,55]. The rapid and short developmental process of the fruit apparently occurs at the expense of current photosynthate resources [56] restricted by the short growth period of the *C. spinosa*.

The average pericarp soluble sugar and starch content of the caper berries developed in the warmth and the sunlight of the coastal sites was substantially higher than that of the fruits from the inland sites. *C. spinosa* exhibits elevated photosynthetic rates during the summer [8,21,24] that may support the pericarp with carbohydrates and especially the textured sticky endocarp (Figure 8) where numerous seeds are adhered [57]; this may play a substantial role in the mechanism of the seed dispersal of *C. spinosa* [36]. The fruits are developed from the plant ovaries and the ovary wall becomes their pericarp; in this context, the sugar budget of the plant tissues plays a regulatory role in both flowering and fruiting. The starch content of the pericarps was neither positively nor negatively correlated to the soluble sugar content of the samples from the coastal and inland sites, implying that during the life span of the fruit (approximately one month) the synthesized sugars are sufficient for metabolic activities.

The average free proline accumulation in the pericarp of the fruits harvested from the coastal and inland habitats, although slightly higher in comparison to the proline content of the petals and leaves of *C. spinosa* [8,17], is low in comparison with that of other tissues from several perennial plants subjected to similar climatic conditions in the eastern Mediterranean during summer [58,59,60,61]. Actually, proline is a key osmolyte contributing to osmotic adjustment, which is involved in stress adaptation and avoidance mechanisms to counteract the loss of turgor. Thus, the relatively low free proline accumulation (Figure 6) is consistent with earlier results on free proline accumulation in other plant tissues of *C. spinosa* [7,8,17], arguing that this species is not subjected to any water deficit conditions during the dry summer period, due to the fact that water can be absorbed via its elongated roots from deep soil layers [13,20]. The average pericarp nitrogen content was higher in the samples from inland sites in comparison with that from coastal sites, with one exception: the value from the Cy coastal site that can be grouped together with those of the inland sites (Figure 5). In other words, the free proline accumulation and nitrogen content in the caper pericarps from the Cy site were significantly different than those of the other coastal sites, which is an interesting result; it is worth mentioning that the Cy site corresponds to Cythera Island off Cape Malea and is located close to inland sites of the Peloponnese peninsula (Figure 9), where numerous caper shrubs develop every year [62].

Plants’ adaptation to environmental stresses is associated with metabolic adjustments that lead to the accumulation of compatible organic solutes (e.g., soluble sugars and proline) that ameliorate the damaging effects of abiotic stress [63,64,65]. The soluble sugars of the pericarp of caper berries were substantially higher than their proline content. The observed low concentrations of proline (which are not indicative of water-stressed tissues) and the negative correlation between proline and soluble sugars (r_s_ = −0.74) probably indicate that the elevated concentrations of sugars are not due to water stress, reinforcing their possible indirect importance in seed dispersal [7,13,20,23,63,64,65,66,67].

It seems likely that proline may contribute to the osmotic adjustment of the fruit pericarp of capers growing in inland habitats, while the substantial soluble sugar content may contribute to the osmotic adjustment of the fruit pericarp of capers growing in coastal habitats, where the glycophyte *C. spinosa* has to cope with abiotic stress. In samples from coastal habitats, the increase in sugar content along a latitudinal gradient coincided with a decline in the nitrogen content (r_s_ = −0.77). Also, a co-transition trend was observed in studied functional traits (soluble sugars and proline) between pericarps and pedicels, which indicates transport continuity.

Given the rapid growth and the short lifespan of the caper fruit during the dry summer in the eastern Mediterranean, the caper’s pericarp composition ascribed to coastal and inland habitats provides valuable information linked to the adaptive process and yield of the *C. spinosa* upon exposure to the considered ambient conditions. Also, the soluble sugars, starch and nitrogen in the pericarp of the *C. spinosa* may be crucial starter substrates for the microbial decay and decomposition of the indehiscent pericarp, facilitating seed germination in natural habitats; further work will be required to fully test this hypothesis.

## 4. Materials and Methods

### 4.1. Research Sites and Samples

The study was conducted on naturally occurring stands of wild plants of *C. spinosa.* The mature fruits of the caper were harvested early in the morning, from 24 July to 24 August (Table 2) from five coastal and three inland habitats in Greece (Figure 9), with a similar topographic relief. The total number of caper fruit samples from observation units corresponded to 18 collections from inland locations and 20 collections from coastal locations. Analytically and according to Table 2, the inland cases were 18: i.e., adding 9 from the University Campus (EK1–EK6 and EK7–EK9, on the 5 and 20 August, respectively), 6 from Kyparissia (Ky1–Ky3 and Ky4–Ky6, on the 16 and 24 August) and 3 from Mets (M1–M3, on the 18 August). Also, according to Table 2, the coastal cases were 20: i.e., adding 3 from Sifnos (S1–S3, on the 25 July), 3 from Cythera (Cy1–Cy3, on the 28 July), 4 from Vouliagmeni (V1–V4, on different dates), 7 from Kea (K1–K7, on different dates) and 3 from Northern Euboea (E1–E3, on the 21 August). In each location, 3–9 plants were selected for the collections and each collection included 15–20 randomly collected specimens of mature fruits.

Mature fruits were collected at the same development stage, same altitude and stem orientation, and were transported in zipper-sealed plastic bags to the laboratory. The pericarps of mature fruits (Figure 1D) with a mean diameter approximately 27–30 mm were used. The small seeds of the fully ripened, sampled fruits (Figure 8) were manually and carefully removed from the pulpy endocarp. The pedicels were separated from the fruits before the measurements. Then, pericarps and pedicels were oven-dried at 60 °C for 72 h [60], ground using a Thomas Wiley Model 4 Mill (Thomas Scientific Swedesboro, NJ, USA) and stored in glass jars in the dark at room temperature for subsequent analyses.

In considering that *C. spinosa* is a widely distributed species throughout the habitats of tropical and subtropical regions and naturalized in the Mediterranean Basin [2,6,68,69], the sampling sites are cited according to increasing latitude (Table 2). A latitudinal diversity gradient is a striking ecological pattern linked to species diversity, which has fascinated biologists [70], as well as the diversity of a *taxon* through several, different habitats [71,72].

The climate in the studied sites is Mediterranean with a marked seasonality, typified by a hot and dry summer season; mean and maximum air temperature and rainfall during the harvesting period, are presented in Table 2.

### 4.2. Determination of Total Soluble Sugars and Starch

Soluble sugars were extracted from dry, finely powdered pericarp and pedicel samples, which were placed in 10 mL 80% ethanol (*v*/*v*) in a shaker and the extracts were filtered using Whatman # 2 filter paper. Soluble sugar concentration was investigated according to a modified phenol-sulphuric acid method [73], using a spectrophotometer (Novaspec III^+^ Spectrophotometer, Biochrom, Cambridge, UK). The determination of starch was made in the residue after the extraction of sugars, using the anthrone method [60]. D-glucose (Serva, Heidelberg, Germany) aqueous solutions were used for the standard curve.

### 4.3. Determination of Nitrogen and Proline

The nitrogen content of the pericarps and pedicels was measured in powdered samples digested with 96% sulphuric acid using a modified Kjeldahl method [61]; a Vapodest 30 distillation system (Gerhardt, Bonn, Germany) was utilized to bind nitrogen with 0.1 N H_2_SO_4_ (Tritisol, Merck, Darmstadt, Germany), and total N was estimated by titration with 0.1 N NaOH (Tritisol, Merck, Darmstadt, Germany). Free proline content was determined colorimetrically on 4 mL samples of the condensed fluid extracted from the finely powdered plant material [74,75]. The extraction procedure from plant samples (pericarps and pedicels) and colorimetric determination were carried out as analytically published [61,73]. Dried, powdered samples were homogenized with aqueous sulphosalicylic acid (20 mL, 3% *w*/*v*), and the homogenate filtered through Whatman # 2 filter paper; 2 mL of the filtrate reacted with acid-ninhydrin solution (2 mL) and glacial acetic acid (2 mL) in test tubes, which were placed in a water bath at 100 °C for 1 h and the reaction terminated in an ice bath. After cooling, the reaction mixture was extracted with 4 mL toluene and homogenized in a vortex. The chromophore containing the toluene was aspirated from the aqueous phase and the absorbance was read at 520 nm, immediately after the terminated reaction in glass tubes placed in an ice bath, using toluene as a blank sample and the spectrophotometer mentioned in paragraph 4.2. Proline concentration was estimated using a standard curve of relevant L-proline solutions (Serva, Heidelberg, Germany) and calculated on a dry weight basis.

### 4.4. Statistical Analysis

The statistical analysis was carried out using R-studio 1.0.136/R3.3.4 [76]. The total number of samples was rather small and measurements did not follow normal distribution; therefore, non-parametric tests were used for statistical analysis. In particular, Spearman correlation coefficient was used to evaluate correlations among parameters and the coefficient (r_s_) is presented in the text. The Kruskal–Wallis test, a non-parametric alternative to one-way ANOVA test, was performed to compare the considered measurements among the sampling sites using the dplyr package [77]. Multiple comparisons were carried out among sampling sites using the Post-Hoc Benjamini and Hochberg adjustment, by using FSA and companion packages [78]. Principal Component Analysis (PCA) of the mean centered capers’ pericarp samples was performed to detect any grouping of the specimens on the basis of the considered eco-physiological traits, using “ade4” and “adegraphics” packages [79,80]. The data were centered in order to normalize the variables so that the considered ecophysiological traits were at the same scale to perform the visualization in a comprehensible way.

## 5. Conclusions

The fruit pericarp of *C. spinosa* plants growing in the wild and collected from coastal Mediterranean habitats contained higher concentrations of soluble sugars and starch than those collected from inland habitats, while the opposite held true for free proline accumulation and nitrogen content; these findings were associated with the relevant pedicel results. The presented functional traits of pericarps and pedicels help further our understanding of the development of this species during the driest period of the year in the Mediterranean region, and complete the previously published work on *C. spinosa*.

## Figures and Tables

**Figure 1 plants-11-03085-f001:**
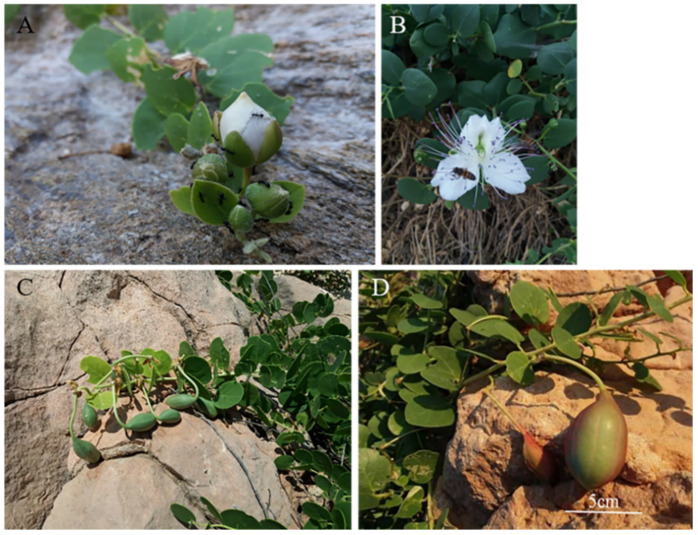
Representative views of the vegetative and reproductive growth of *Capparis spinosa*: (**A**) Stem with expanded and expanding leaves, and flower buds. (**B**) Flowering time at dusk; a range of floral traits including size, symmetry, odor, color and a pair of variegated petals with white marginal and green basal petal portions angled so as to face the sunrise, determine pollinator visitation in order to access reward from flower [18] of *C. spinosa*. (**C**) Stems, leaves and successive green immature fruits with pedicles. (**D**) Stem, leaves and a mature fruit—brownish and greenish—with pedicel, next to a detached and dehydrated substantially smaller fruit.

**Figure 2 plants-11-03085-f002:**
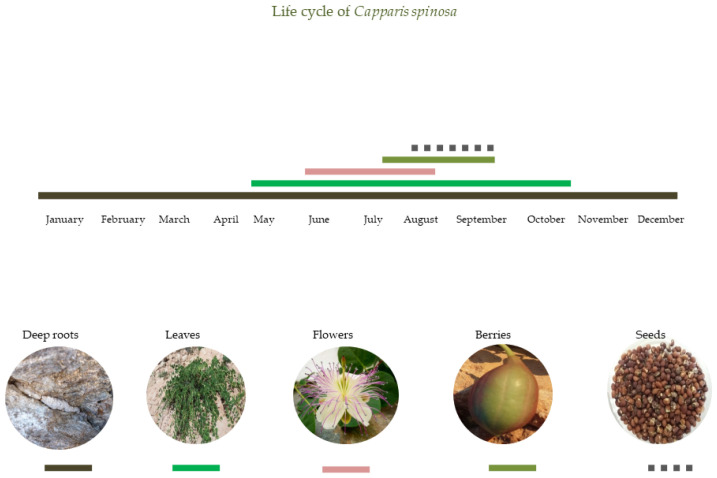
Schematic presentation of the life cycle and phenological stages (indicated using colorful horizontal lines and a dotted line) of *C. spinosa*: leaf initiation and longevity (green line) from May to October, flowering (pinkish line) from June to mid-August, fruiting (olive-green line) from late July to mid-September, seed maturation (dark dotted line) from late August to September, and roots existing throughout the year.

**Figure 3 plants-11-03085-f003:**
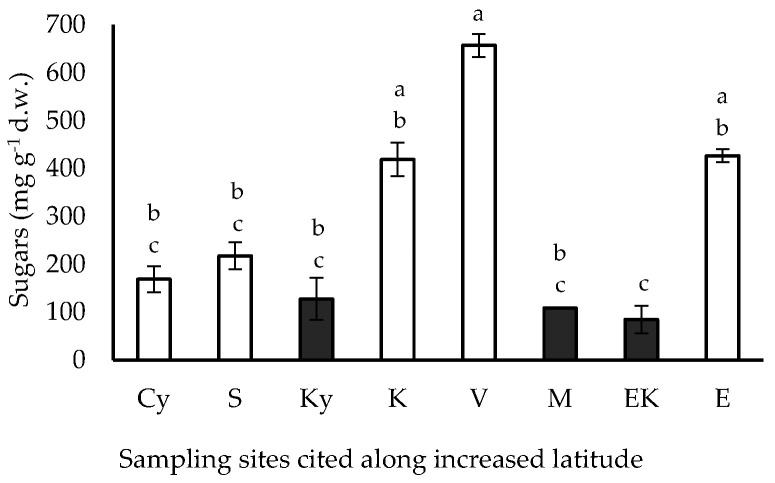
Soluble sugar content in the fruit pericarp of *C. spinosa* harvested from inland (closed bars) and coastal habitats (open bars), cited along increased latitude. The values are means ± standard deviation; significant difference (*p* < 0.05) was detected using Kruskall–Wallis’ test and is marked using lowercase letters, according to post hoc analysis.

**Figure 4 plants-11-03085-f004:**
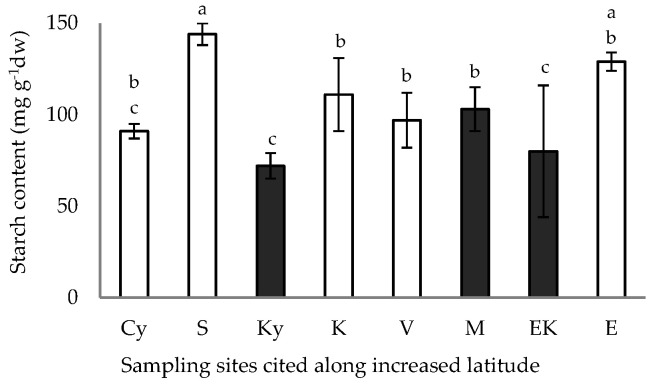
Starch content of the fruit pericarp of *C. spinosa* harvested from inland (closed bars) and coastal habitats (open bars) cited along increased latitude. The values are means ± standard deviation; significant difference (*p* < 0.05) of mean values was detected using Kruskall–Wallis’ test and is marked using lowercase letters, according to post hoc analysis.

**Figure 5 plants-11-03085-f005:**
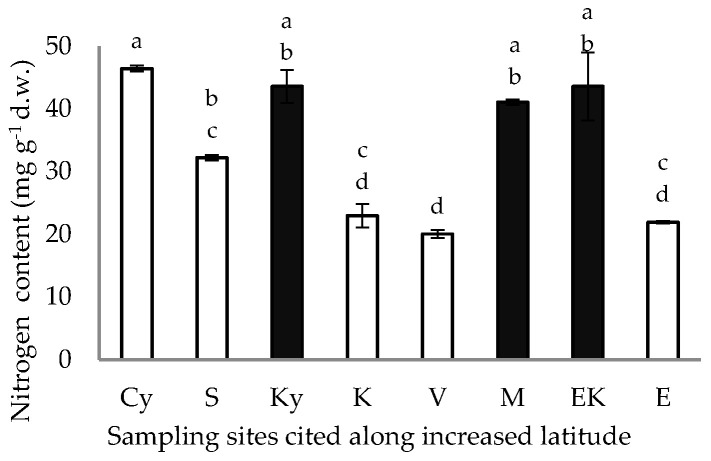
Nitrogen content of the fruit pericarp of *C. spinosa* harvested from inland (closed bars) and coastal habitats (open bars), cited along increased latitude. The values are means ± standard deviation; significant difference (*p* < 0.05) of mean values was detected using Kruskall–Wallis’ test and is marked using lowercase letters, according to post hoc analysis.

**Figure 6 plants-11-03085-f006:**
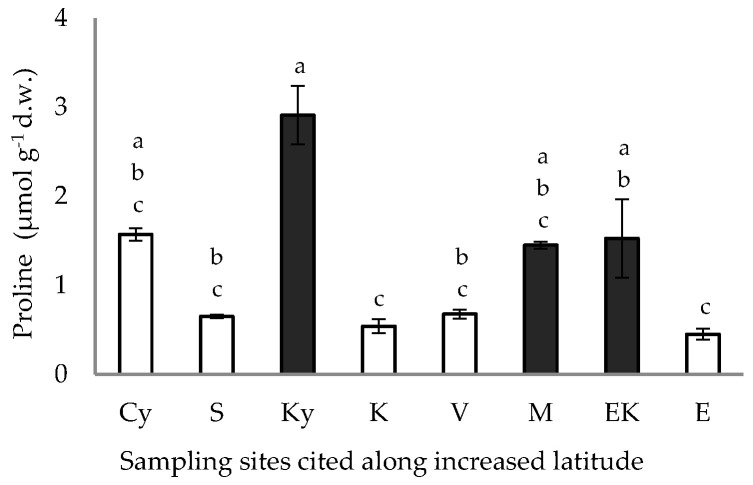
Free proline accumulation in fruit pericarp of *C. spinosa* harvested from inland (closed bars) and coastal habitats (open bars), cited along increased latitudinal gradient. The values are means ± standard deviation; significant difference (*p* < 0.05) of mean values was detected using Kruskall–Wallis’ test and is marked using lowercase letters, according to post hoc analysis.

**Figure 7 plants-11-03085-f007:**
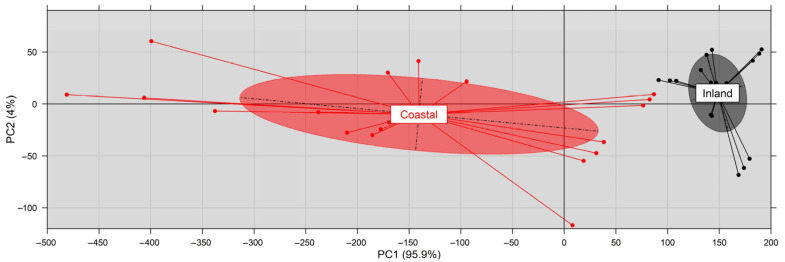
Visualization of PCA based on the considered functional traits of caper pericarps from inland and coastal sites, revealing the grouping of specimens derived from coastal (red) versus inland (grey) habitats.

**Figure 8 plants-11-03085-f008:**
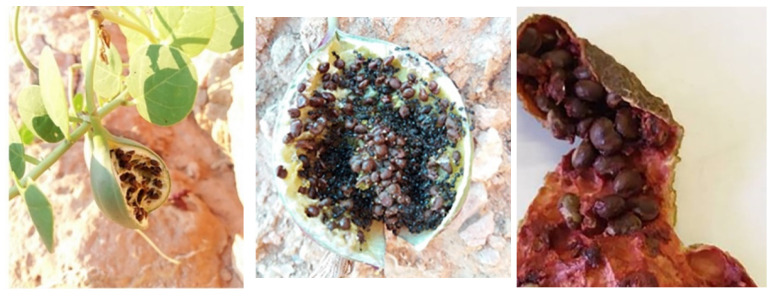
(**Left**) Mature dehiscent fruit of *Capparis spinosa*; (**center**) half of a caper fruit with seeds and ants (magnified ×5); (**right**) part of the sticky fruit endocarp with adhered seeds (magnified ×20).

**Figure 9 plants-11-03085-f009:**
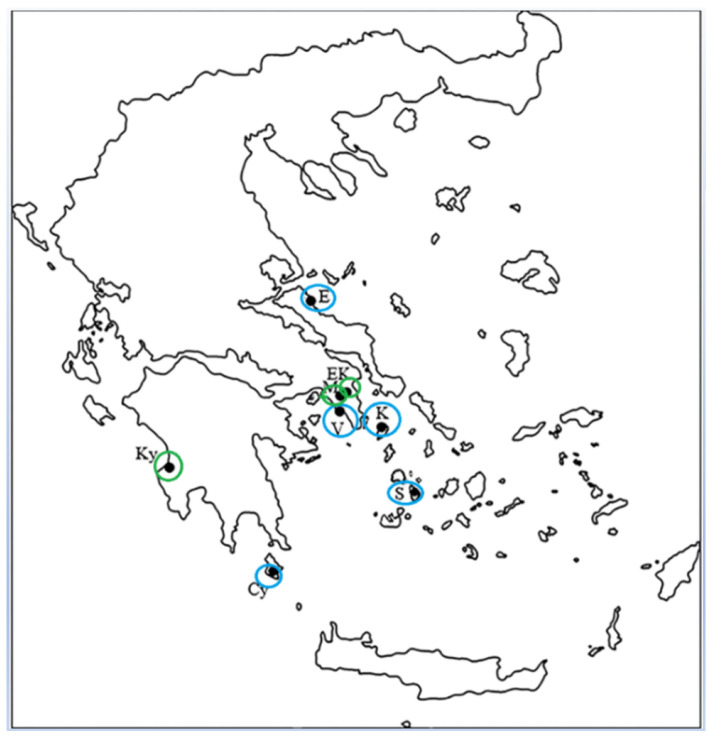
Abstract map of the territory of Greece, where the coastal (blue circles) and inland (green circles) sampling sites are indicated using initials (see Table 2).

**Table 1 plants-11-03085-t001:** Functional traits in pedicels of wild caper fruits. Values are means of five replicates ± standard deviation; significant differences (*p* < 0.05) are marked using lowercase superscript letters that are given separately on each column variable. Information about habitat’s initial is given in Table 2.

Initial of Habitat Cited Along Increased Latitude	Soluble Sugars (mg g^−1^)	Starch (mg g^−1^)	Proline (μmol g^−1^)	Nitrogen (mg g^−1^)
Cy	150 ± 3.0 ^d^	89 ± 0.8 ^a^	0.42 ± 0.011 ^a^	23.80 ± 0.20 ^a^
S	125 ± 4.2 ^e^	74 ± 1.3 ^b^	0.16 ± 0.015 ^d^	11.20 ± 0.12 ^e^
Ky	107 ± 2.1 ^f^	50 ± 0.6 ^d^	0.33 ± 0.014 ^b^	15.90 ± 0.43 ^c^
K	184 ± 2.5 ^c^	58 ± 0.7 ^c^	0.20 ± 0.085 ^c^	15.32 ± 0.65 ^c^
V	265 ± 7.5 ^a^	31 ± 0.3 ^e^	0.30 ± 0.025 ^b^	14.43 ± 0.37 ^d^
M	125 ± 3.4 ^e^	88 ± 1.0 ^a^	0.17 ± 0.004 ^d^	15.40 ± 0.47 ^c^
EK	129 ± 1.8 ^e^	51 ± 0.5 ^d^	0.30 ± 0.014 ^b^	17.76 ± 0.50 ^b^
E	244 ± 6.0 ^b^	38 ± 2.0 ^e^	0.22 ± 0.003 ^c^	10.85 ± 0.23 ^e^

**Table 2 plants-11-03085-t002:** The sampling sites cited according to the succession of the Julian day of sampling, with geographic and climatic parameters, and sample’s indication.

Julian Day of Harvesting	Names of Sampling Sites (Initials)	Longitude	Latitude	Mean and Maxiumn Temperature (°C)	Rainfall (mm)	Sample Indication
24 July	Vouliagmeni (V)	23.7736° E	37.7983° N	28.2 and 32.2	0.7	V1
25 July	Sifnos (S)	24.6607° E	37.0612° N	25.1 and 27.1	0.4	S1–S3
27 July	Vouliagmeni (V)	23.7736° E	37.7983° N	28.3 and 32.4	0.6	V2
27 July	Kea (K)	24.3146° E	37.6415° N	25.6 and 27.9	0.9	K1
28 July	Cythera (Cy)	22.9904° E	36.2633° N	26.0 and 29.1	0.3	Cy1–Cy3
5 August	Univ. Campus (EK)	23.7870° E	37.9664° N	28.4 and 32.9	0.3	EK1–EK6
6 August	Vouliagmeni (V)	23.7736° E	37.7983° N	28.2 and 32.5	0.2	V3
7 August	Kea (K)	24.3146° E	37.6415° N	25.3 and 27.6	0.0	K2
10 August				25.3 and 27.5	0.0	K3
11 August				26.2 and 28.3	0.2	K4
12 August				26.2 and 28.4	0.8	K5
13 August				26.2 and 28.4	0.0	K6
14 August	Vouliagmeni (V)	23.7736° E	37.7983° N	29.3 and 33.2	0.0	V4
16 August	Kyparissia (Ky)	21.6779° E	37.2488° N	26.2 and 31.4	1.4	Ky1–Ky3
18 August	Mets (M)	23.7361° E	37.9638° N	27.4 and 32.6	3.7	M1–M3
20 August	Univ. Campus (EK)	23.7870° E	37.9664° N	26.6 and 31.9	3.5	EK7–EK9
21 August	Northen Euboea (E)	23.4432° E	38.8841° N	26.3 and 31.2	7.0	E1–E3
22 August	Kea (K)	24.3146° E	37.6415° N	22.2 and 24.7	5.4	K6
24 August	Kyparissia (Ky)	21.6779° E	37.2488° N	22.8 and 28.4	6.7	Ky4–Ky6

## Data Availability

The data are available from the authors upon request.

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
