# Peer review of "Comparison of Pericarp Functional Traits in Capparis spinosa from Coastal and Inland Mediterranean Habitats"

_plants, 2022, doi:10.3390/plants11223085_

Round 1

Reviewer 1 Report

‘Comparison of Pericarp Functional Traits in Capparis spinose from Coastal and Inland Mediterranean Habitats’ - review

The material and methods section is at the end of the manuscript. It is very uncommon situation. When I read the results, I didn’t fully understand (and check) because I didn’t know the used methods. Is it really managed by the journal?

I have some major comments on the methodology. The statistical analysis was described very briefly thus I’m not sure if my corrects correspond to the author's' intention. I have serious doubts, almost certain that the PCA analysis and the regression analysis was wrongly done.

Major comments - M&M section

‘The study was conducted on naturally occurring stands of wild plants of C. spinosa’ – So, the effect of location is inseparably connected with genotypes composition occurring in each of the location. I think that this can be omitted if the number of plantations were high, but there were only a few plantations. Moreover I suppose that the regression was done also for 2 subsets (coastal/inland) what decreased number of observations.

Probably the mentioned regression was wrong – for inland there were only 3 cases (observation units = locations), so this is extremely less than usually recommended 30-40. If the regression was made based on single fruits, then the assumption of regression was not met (independence of observation units)! Thus I suggest to drop of this regression analysis. In this case the covariance analysis (such a combination of ANOVA and regression) should be done.

PCA – I think that the authors make a mistake. They use ‘factoextra’ package and they didn’t manage to don’t use centering (that is with default options). Moreover centering was made across traits for each observation, not across observations – what is fault for me. The PCA must be described with details, and if some transformations was used, then they should be described (maybe with the reason).

 ‘in each sampling site five caper bushes were selected at random and each sample indication corresponds to 15-20 randomly collected fruit’ and ‘Determination of total soluble sugars and […]’ – were the fruit coming from the same caper bush were homogenized or was each fruit analyzed separately? this difference affects the type of statistical analysis that will be appropriate for the data set.

It is not clear how the regression analysis was performed. What were replications cases (that is, the observation units)? I found only ‘Soluble sugars were extracted from dry, finely powdered pericarp and pedicel samples’ but I don’t know if the samples coming from one fruit, from homogenized fruit from one plantation, or from subset of fruit from one plantation. What were dependent variables and independent variables? There were lack of analysis of variance tables for regression analyses.

Minor comments:

Fig .3. caption: ‘significant difference (p < 0.05) is marked using lowercase letters’  - could you write the name of the test?

Fig.3. I suppose that ‘standard error’ means standard error of the mean. And it is very strange to me that bar V (lowercase a) is not significantly higher than bar S (lowercase ab). Please check it. The same situation occurred in fig. 5 – the SME close to 0 means  that differed bars (for example S and E) should significant differ but by lettering it is not so.

‘2.2. Soluble sugars ‘ section, ‘The highest sugar content was detected in V (Figure 9)’ – but not significantly higher than in 4 other cases! ‘lowest in EK’ the same comment.

‘(Figure 9)’ was firs time cited after fig. 3, before figure 4. The fig. 9 should be in materials section, and then it will be before this part of the text.

 ‘2.5. Free Proline accumulation’ section – ‘as lower than that of the inland sites’ – based on the homogeneity the locations: S and V were not significantly lower than M and EK. Thus this sentence was not based on a statistically significant fact.

‘2.6. Principal Component Analysis’ section – The PCA is not well described in the M&M section. The ‘Table 1. Principal component analysis (PCA)’ suggest (for me) that the data set was centered for each caper bush across variables – and this is not appropriate. Please describe the PCA with details, such as centering, scaling.

The fig. 7. indicates that there were different number of bushes in locations – from 3 to 9. In the same time ‘in each sampling site five caper bushes were selected at random’ – what is the truth?

‘2.7. Pedicels’ section – ‘A negative relationship was obtained between soluble sugars and starch of pedicels from coastal sites (y= –0.376x+130.72, R2=0.862, p<0.05)’ – This is confusing. ‘relationship between’ is associated with the correlation coefficient whereas the regression function was put in the parenthesis. Moreover it is not wrote what is x and what is y. The next, why x influenced on the y and not in inverse? This is regression function (probably standard simple regression, so the equation can’t be invert as could be done for mathematical equation). Maybe such parenthesis is enough (r=-0.928, at p=__)? For other traits in this paragraph I can make the same comment. Of course, if authors will use regression function, it should be defined what is independent variable (and why) and what is the dependent variable. And in this case the current parentheses are OK

I did not find information about when the experiment was carried out. Was it in one year only? If it was a multi-years then this fact affect the statistical analysis.

Author Response

the response to the Review report (Reviewer 1 )  is submitted below as an attached file.

Reviewer 2 Report

The paper reports the investigation of some functional traits of the pericarp of Capparis spinosa fruit in coastal and inland of Greece. The paper is well written, and it fills a gap in the chemical and physiological knowledge of this product. However, the authors should improve some aspects of the manuscript such as the aim and the reason why they decided to consider only the analyzed parameters.

Abstract

“Higher sugar and starch content was investigated in the pericarps and fruit-pedicels harvested from wild caper plants grown in coastal habitats in comparison with those from inland habitats, while the opposite holds true for proline and nitrogen content.”

Is not clear what the Authors mean in the second part of the sentence.

Introduction

The Authors should better clarify the aim of the work. Why is it important to investigate these parameters? I understood that they have never been studied but understanding the motivation that pushes the Authors to study them is not clear at this stage.

Material and Methods

Why were samples dried? Wasn't it better to analyze the fresh or frozen product? At which temperature were the samples dried?

Conclusions

“these findings were accosiated by the relevant to pedicels’ results”

Is not clear what the Authors mean.

Author Response

the response to the comments of Reviewer 2 is submitted below as an attached file.

Round 2

Reviewer 1 Report

Previous comment: PCA – I think that the authors make a mistake. They use ‘factoextra’ package and they didn’t manage to don’t use centering (that is with default options). Moreover centering was made across traits for each observation, not across observations – what is fault for me. The PCA must be described with details, and if some transformations was used, then they should be described (maybe with the reason).

Authors response: PCA has been replaced; in fact, PCA was performed again using “ade4” and “adegraphics” packages. The data were centered in an appropriate way to normalize the variables so that all features are at the same scale in order to perform the visualization in a comprehensible way; with the current analysis the separate grouping between coastal and inland caper fruits on the basis of the recorded ecophysiological traits is shown.

Comment: thank you. The text ‘The data were centered’ could be put into the method section. It should be check, if you only centered variables – because you have variables of different types, probably the variables were standardized. Thus the standardization of variables should be mentioned in the methods section.

Previous comment: Fig.3. I suppose that ‘standard error’ means standard error of the mean. And it is very strange to me that bar V (lowercase a) is not significantly higher than bar S (lowercase ab). Please check it. The same situation occurred in fig. 5 – the SME close to 0 means that differed bars (for example S and E) should significant differ but by lettering it is not so. 3

Authors response: it has been corrected.

Comment: but it is not clear for me what means the lines up and down the top of the bars. Maybe it is standard deviations (instead the standard error of the mean)? If yes, please change it.

The same is in the table 1: ‘Values are means of five replicates ± SE;’. We can read that ‘Cy = 150 ± 3.0b’ and ‘K = 184 ± 2.5a,b’ so 150+3 is strictly lower 184-2.5. Thus the groups should be insignificant differentiated. It should be carefully checked and explained, and proper description placed with charts and tables

Author Response

Our reply to the review report is in the attached file.
